# OpenReview forum: "CounselBench: A Large-Scale Expert Evaluation and Adversarial Benchmarking of Large Language Models in Mental Health Question Answering"
_ICLR.cc/2026/Conference — ICLR 2026 Oral_

### Official Review · Reviewer_eiyu · 2025-10-30

**Soundness:** 4
**Presentation:** 4
**Contribution:** 4
**Rating:** 8
**Confidence:** 5

**Summary:**

This paper introduces a new benchmark for evaluating LLMs on open-ended mental health question answering (QA) questions. The questions are sourced from 100 real, patient-facing mental health questions posted on CounselChat forum. 100 licensed mental health professionals rated responses from 3 LLMs and the original human therapist response and evaluated across 6 dimensions, quality, empathy, specificity, medical advice, factual consistency, toxicity. The expert annotations include not only numeric ratings but contain span-level evidence extraction and written rationales. An additional dataset is created by 10 of the experts designed to trigger specific failure modes identified from the benchmark evaluation.

**Strengths:**

* The benchmark is sourced from a forum where responses were provided prior to 2022 and is highly likely from a trained human therapist and is co-designed with mental health professionals. The six evaluation dimensions are clinically grounded in clinical psychology literature.
* The benchmark contains evaluation from 100 licensed mental health professionals encompassing 32 license types and 43 specialization areas. There is rich annotations and represents one of the largest expert-annotated datasets for open-ended mental health QA but also has high interannotator agreement.
* There are two benchmarks that covers not only systematic evaluation of the LLM and human responses, but an expert-authored adversarial set designed to stress-test LLMs.
* Provides a critical evaluation of LLM-as-Judge paradigm highlighting the inability to flag problematic content and safety concerns identified by human experts.
* Multiple pilot studies were introduced to refine the evaluation protocol with thorough analysis of annotator behavior and clear methodology provided.

**Weaknesses:**

* There are slight scale limitations of the dataset as there are only 100 questions (5 per topic, 20 topics) which were chosen by popularity and may not fully reflect the diversity and subtly of mental health questions (as motivated by in the introduction). Similarly, the adversarial benchmark contains only 120 questions, although there are 20 per failure mode.
* There is a heavy reliance on automated evaluation of the adversarial responses which seems to go against the findings of the LLM-as-Judge critique. The same model was also used to evaluate the failure modes that were identified in the original responses of the first benchmark (GPT-4.1 and GPT 4, respectively).
* The adversarial benchmark, while interesting from a high-level perspective, is confusing. It seems to determine whether high trigger rates indicate model problems as determined by GPT-4.1. However, it's not clear what makes a question adversarial versus difficult? Also, it has been designed from 2 failure modes from the 3 original base models but then tested across new models (like Claude). Is the expectation that they are model-specific problems or really universal vulnerabilities?
* Is it truly fair to benchmark against the human baselines as they were often informal and vary in quality as they could be answered anonymously. Are these fair comparisons with one another?

Minor:
* The use of 2000 expert annotations is slightly misleading as the dataset contains 100 total questions with 4 different responses and 5 annotators per response pair. This is never explicitly stated as this breakdown.
* Phrasing in line 404-405 is slightly confusing?
*There are inconsistent periods throughout the text. For example, line 182 doesn't have an ending period and the different paragraph headings sometimes have periods and sometimes don't (e.g., line 269, 313, etc.).

**Questions:**

1. With only 100 questions (5 per topic) selected primarily by popularity (upvotes), does this benchmark captures the diversity and subtlety of mental health questions?
2. What makes the questions in the COUNSELBENCH-ADV adversarial in nature? Is it mostly in difficulty or along other dimensions?
3. The main results on COUNSELBENCH-EVAL is that LLM as judges might miss safety concerns yet the adversarial responses are then predominately judged using GPT-4.1, with only ~3% manually validated with a 72.9% agreement with human experts. This seems to be at odds with one another, and a bit hard to determine what might be the core takeaway.
4. Adversarial questions are designed from 3 specific models, but the results does not suggest that these are universal vulnerabilities. Should we expect a question designed to trigger specific vulnerabilities to also trigger issues across model families?

---

> ### Author Response · Authors · 2025-11-21
> **Response to W1 and Q1**
>
> Thank you very much for your helpful feedback and encouragement! We respond to each point below:
>
> 1. (W1+Q1) **W1**: There are slight scale limitations of the dataset as there are only 100 questions (5 per topic, 20 topics) which were chosen by popularity and may not fully reflect the diversity and subtly of mental health questions (as motivated by in the introduction). Similarly, the adversarial benchmark contains only 120 questions, although there are 20 per failure mode. | **Q1**: With only 100 questions (5 per topic) selected primarily by popularity (upvotes), does this benchmark captures the diversity and subtlety of mental health questions?
>
> Thank you for this comment. First, we clarify that evaluation is conducted at the QA-pair level, meaning each response to the same question requires an independent expert assessment. CounselBench-Eval contains 400 distinct QA pairs (100 questions × 4 responses), and 2,000 different expert annotations, each with its own written rationale and span-level labels. This scale is comparable to prior expert-annotated mental-health datasets. For example, MentalQA [1] includes 500 QA pairs annotated by 3 raters for two classification tasks (question type and answer strategy), and  MENTAT [2] evaluates the quality of answer options for mental-health, exam-style multiple-choice question and collected 657 annotations on 57 questions, where 8 clinicians rate each answer option on a 0–100 quality scale. In contrast, we have 400 free-form QA pairs evaluated by five different annotators (100 annotators and 2000 annotations in total).
>
> Our rich annotation protocol substantially increases information per instance. Each QA pair receives six evaluation dimensions, span-level labels, and written rationales. Annotators spent a median of 4.1 minutes per QA pair (IQR 2.6-9.45), making annotation far slower and more costly than standard categorical labeling. This depth of supervision enables a much more comprehensive evaluation of model behavior. For example, the written rationales support failure-mode analysis and can be further used to train critique models or interpretability tools; the span-level annotation can be used to develop automated quality or safety detectors.
>
> Finally, the questions themselves are diverse. They span 20 different topics, and in the revision we added Table 13 to summarize the keywords for all 100 questions. For example, the five questions in Depression include:
> > Questions in Depression: (1) sad time don’t like family; (2) depressed care busy relationships; (3) husband wants divorce diagnosed; (4) feel self-harm addiction; (5) tell parents depressed need help.
>
> Similar diversity appears across the remaining topics, demonstrating substantial within-topic variation.
>
> [1] H. Alhuzali, A. Alasmari, and H. Alsaleh, “MentalQA: An Annotated Arabic Corpus for Questions and Answers of Mental Healthcare,” arXiv.org, 2024.‌
>
> [2] M. Lamparth et al., “Moving Beyond Medical Exam Questions: A Clinician-Annotated Dataset of Real-World Tasks and Ambiguity in Mental Healthcare,” arXiv.org, 2025.

---

> ### Author Response · Authors · 2025-11-21
> **Response to W2 and Q3**
>
> 2. (W2+Q3) **W2**: There is a heavy reliance on automated evaluation of the adversarial responses which seems to go against the findings of the LLM-as-Judge critique. The same model was also used to evaluate the failure modes that were identified in the original responses of the first benchmark (GPT-4.1 and GPT 4, respectively). | **Q3**: The main results on COUNSELBENCH-EVAL is that LLM as judges might miss safety concerns yet the adversarial responses are then predominately judged using GPT-4.1, with only ~3% manually validated with a 72.9% agreement with human experts. This seems to be at odds with one another, and a bit hard to determine what might be the core takeaway.
>
> Thank you for raising this important concern. We agree that relying on LLM-as-Judge for evaluation would be inconsistent with our findings, and the spot-check agreement is not sufficiently high. Based on your feedback, we made a major change to the evaluation of COUNSELBENCH-ADV.
>
> **We have replaced the primary GPT-4.1 evaluation with full human annotation.** We recruited five mental-health professionals (different from those who authored the adversarial questions) to annotate 1,080 QA pairs (120 adversarial questions × 9 models × 1 response). In this setting, the task is a single binary label (whether the issue is present or not), which is fast and straightforward for experts to evaluate. Details about the evaluation setup are provided in Sec. 5 and Appendix H.
>
> The main evaluation results for COUNSELBENCH-ADV are now entirely based on human expert judgments. The updated Table 3 with expert-annotated failure rates is included in the revision:
> | Specific Issue | GPT-3.5-Turbo | GPT-4 | GPT-5 | Llama-3.1 | Llama-3.3 | Claude-3.5-Sonnet | Claude-3.7-Sonnet | Gemini-1.5-Pro | Gemini-2.0-Flash |
> |----------------|--------------:|------:|------:|----------:|----------:|-------------------:|-------------------:|---------------:|-----------------:|
> | 1. Medication  |          0.05 |  0.00 |  0.47 |      0.05 |      0.10 |               0.00 |               0.00 |           0.00 |             0.00 |
> | 2. Therapy     |          0.20 |  0.20 |  0.85 |      0.55 |      0.65 |               0.45 |               0.50 |           0.20 |             0.26 |
> | 3. Symptoms    |          0.15 |  0.45 |  0.60 |      0.45 |      0.45 |               0.50 |               0.37 |           0.26 |             0.25 |
> | 4. Judgmental  |          0.25 |  0.25 |  0.05 |      0.11 |      0.10 |               0.05 |               0.10 |           0.20 |             0.10 |
> | 5. Apathetic   |          0.70 |  0.20 |  0.15 |      0.15 |      0.15 |               0.05 |               0.20 |           0.40 |             0.30 |
> | 6. Assumptions |          0.40 |  0.35 |  0.15 |      0.25 |      0.25 |               0.35 |               0.25 |           0.40 |             0.35 |
>
> We further systematically analyze the performance of LLMs as judges for identifying failure modes. We added Table 4, which reports Accuracy and F1 of LLM judges relative to human experts:
>
> | Metrics | GPT-3.5-turbo | GPT-4 | GPT-4.1 | GPT-5 | Llama-3.1-70B-Instruct | Llama-3.3-70B-Instruct | Claude-3.7-sonnet | Gemini-2.0-flash |
> |---------|---------------:|------:|--------:|------:|-----------------------:|-----------------------:|------------------:|-----------------:|
> | Acc.    |           0.74 |  0.70 |    0.64 |  0.67 |                  0.63  |                  0.64  |              0.70 |             0.63 |
> | F1      |           0.41 |  0.35 |    0.49 |  0.49 |                  0.48  |                  0.48  |              0.50 |             0.46 |
>
> (*Gemini-1.5-Pro and Claude-3.5-Sonnet, which were included in the LLM-as-judge analysis in CounselBench-EVAL, were not longer available when we ran the LLM-as-judge experiments for CounselBench-ADV)
>
> We again observe a substantial gap between LLM judgments and human experts on COUNSELBENCH-ADV. The performance breakdown of each LLM judge for each failure mode is present in Figure 8. This reinforces that LLM-as-Judge is unreliable in this domain and highlights the importance of the fully human-annotated results now presented.
>
> We have incorporated these changes and the full updated analysis in Sec. 5 of the revision.

---

> ### Author Response · Authors · 2025-11-21
> **Response to W3, Q2 and Q4**
>
> 3. (W3+Q2+Q4) **W3**: The adversarial benchmark, while interesting from a high-level perspective, is confusing. It seems to determine whether high trigger rates indicate model problems as determined by GPT-4.1. However, it's not clear what makes a question adversarial versus difficult? Also, it has been designed from 2 failure modes from the 3 original base models but then tested across new models (like Claude). Is the expectation that they are model-specific problems or really universal vulnerabilities? | **Q2**: What makes the questions in the COUNSELBENCH-ADV adversarial in nature? Is it mostly in difficulty or along other dimensions? | **Q4**: Adversarial questions are designed from 3 specific models, but the results does not suggest that these are universal vulnerabilities. Should we expect a question designed to trigger specific vulnerabilities to also trigger issues across model families?
>
> Thank you for these helpful questions. We have added more clarification of the construction of COUNSELBENCH-ADV in the revision.
>
> First, the goal of COUNSELBENCH-ADV is not to make questions “difficult”, but to recreate realistic questions that reliably elicit specific failure modes observed in COUNSELBENCH-EVAL. To make this explicit, we have added more details to the revision and included the full questionnaire used by clinicians in Figure 5 of Appendix A. Annotators were shown concrete failure cases and instructed to author new questions that would plausibly trigger the same failure mode, without increasing complexity or introducing artificial difficulty.
>
> Second, our method follows established approaches in adversarial data generation. Prior work commonly begins with seed adversarial examples, and then amplifies [1,2], imitates [3,4], or perturbs [5] them to produce new adversarial instances. Our approach parallels this paradigm but replaces synthetic LLM-generated adversaries with clinician-authored ones. This avoids artifacts introduced by synthetic prompts and ensures that the resulting questions remain realistic and clinically grounded. Our examples can also be used as seeds for future synthetic extension.
>
> Finally, we clarify three aspects regarding whether these failure modes are model-specific: (1) The six failure modes capture behaviors such as judgmental tone, dismissiveness, unwarranted symptom speculation, and unconstructive feedback, all of which are broadly inappropriate in therapeutic communication and are well-documented in counseling psychology literature [6, 7]. These are not tied to any particular architecture.
> (2) Although the failure modes were derived from three base models, we observe similar vulnerabilities across multiple model families in COUNSELBENCH-ADV. For example, Claude is not included in COUNSELBENCH-EVAL, so it does not have adversarial examples, but it still shows substantial rates of therapy-suggestion, symptom-speculation, and assumption-based errors (updated Table 3). This pattern appears across GPT, Gemini, and LLaMA as well, indicating that the adversarial prompts reveal shared vulnerabilities rather than model-specific flaws.
> (3) The idea of constructing universally adversarial questions is interesting and feasible. One possibility is to present professionals with failure cases from multiple models simultaneously, but this would require careful design to address the cognitive load and difficulty of merging multiple cases.
>
> [1] M. Bartolo, T. Thrush, R. Jia, S. Riedel, P. Stenetorp, and D. Kiela, “Improving Question Answering Model Robustness with Synthetic Adversarial Data Generation,” Proceedings of the 2021 Conference on Empirical Methods in Natural Language Processing, Jan. 2021.
>
> [2] L. Jiang et al., “WildTeaming at Scale: From In-the-Wild Jailbreaks to (Adversarially) Safer Language Models,” arXiv.org, 2024.
>
> [3] A. Liu, S. Swayamdipta, N. A. Smith, and Y. Choi, “WANLI: Worker and AI Collaboration for Natural Language Inference Dataset Creation,” arXiv.org, 2022.
>
> [4] Y. K. Lal, Preethi Lahoti, A. Sinha, Y. Qin, and Ananth Balashankar, “Automated Adversarial Discovery for Safety Classifiers,” pp. 13–26, Jan. 2024.
>
> [5] T. Wu, Marco Tulio Ribeiro, J. Heer, and D. S. Weld, “Polyjuice: Generating Counterfactuals for Explaining, Evaluating, and Improving Models,” Jan. 2021.
>
> [6] Counselling Tutor, “Carl rogers’ core conditions,” Counselling Tutor, 2025.
>
> [7] R. for, K. Ernstmeyer, and E. Christman, “Chapter 2 Therapeutic Communication and the Nurse-Client Relationship,” Nih.gov, 2022.

---

> ### Author Response · Authors · 2025-11-21
> **Response to W4, Minor**
>
> 4. (W4) Is it truly fair to benchmark against the human baselines as they were often informal and vary in quality as they could be answered anonymously. Are these fair comparisons with one another?
>
> Thank you for raising this point. First, we want to quickly clarify: CounselChat responses are written by verified, non-anonymous licensed therapists with public profiles; only the patient questions are anonymous. We have added this emphasis in Sec. 4.1 to avoid confusion:
> > All responses on CounselChat are written by verified, non-anonymous therapists with public profiles; only patient questions are anonymous [1].
>
> We conducted a brief quality check of the online therapist responses (median length 142 words, IQR: 95.75-183.5; median 372.5 views, IQR: 248.5-785.5), indicating reasonable substance and engagement.
>
> We fully agree that these online responses are not equivalent to those produced in formal clinical encounters. Due to stringent privacy protections, real patient-provider treatment data and clinically supervised counseling transcripts cannot be released publicly, and therefore cannot be used to build public benchmark datasets. Compared to general peer-support forums like Reddit, CounselChat provides comparatively higher-quality responses from verified clinicians, making it one of the best publicly accessible sources for open-ended mental-health QA. To avoid overstating the comparison, we carefully frame the responders as “online human therapists” throughout the paper (rather than general "human" or “human therapists”).
>
> [1] N. Bertagnolli, “Counsel Chat: Bootstrapping High-Quality Therapy Data,” Medium, May 11, 2020.
>
> -
>
> Minor:
> 1. The use of 2000 expert annotations is slightly misleading as the dataset contains 100 total questions with 4 different responses and 5 annotators per response pair. This is never explicitly stated as this breakdown.
>
> We thank the reviewer for raising this concern. We have clarified this in the revision and now present the full structure in the Introduction:
> > CounselBench-Eval: we recruit 100 licensed or professionally trained mental-health professionals to provide 2,000 evaluations of answers to real patient questions. These evaluations cover 4 answer providers (GPT-4, LLaMA-3, Gemini, and online human therapists) across 100 questions, with 5 independent expert annotations per response. Each annotation includes span-level labels and written rationales.
>
> We have also added the explicit breakdown in Sec. 4.4:
> > In total, the study includes 2,000 QA annotations (100 questions × 4 answers × 5 annotations).
>
>
> 2. Phrasing in line 404-405 is slightly confusing? *There are inconsistent periods throughout the text. For example, line 182 doesn't have an ending period and the different paragraph headings sometimes have periods and sometimes don't (e.g., line 269, 313, etc.).
>
> We thank the reviewer for catching these issues. For lines 404-405, we have revised the sentence to: “To construct a targeted adversarial benchmark, we first need to isolate the failure modes that LLMs exhibit.” We also reviewed and corrected instances where periods were missing.

---

### Official Review · Reviewer_HDvA · 2025-10-30

**Soundness:** 3
**Presentation:** 3
**Contribution:** 3
**Rating:** 6
**Confidence:** 3

**Summary:**

This paper introduces CounselBench, a clinician-grounded benchmark for open-ended mental-health QA composed of two complementary parts:

- CounselBench-Eval: 2,000 expert evaluations produced by 100 licensed or professionally trained mental-health practitioners, who rate and annotate responses (LLMs + human therapists) to 100 real patient questions across six evidence-based dimensions: overall quality, empathy, specificity, factual consistency, medical advice, and toxicity; with span-level evidence and written rationales.
- CounselBench-Adv: an expert-crafted adversarial set of 120 prompts designed to prospectively elicit clinically meaningful failure modes observed in CounselBench-Eval.

The pipeline samples real questions from CounselChat, obtains model and human responses, and collects blind expert ratings (five raters per item), enabling fair comparison and inter-rater reliability analysis.  The authors find that state-of-the-art LLMs can score well on several quality dimensions, yet recur in clinically salient errors. Failure mode distributions exhibit model-family patterns and can shift across major releases.

**Strengths:**

- **High-Quality Evaluation and Scale**: The paper's most significant contribution is the rigor and scale of its evaluation. It did not rely on students or crowdworkers, but instead recruited 100 mental health professionals to provide 2,000 evaluations. These 100 professionals hold 32 distinct license/degree types and span 43 specialization areas, ensuring the professionalism and diversity of the results.
- **Clinically Grounded Evaluation Dimensions**: The study moves beyond simple "accuracy" to define six clinically crucial dimensions: overall quality, empathy, specificity, factual consistency, medical advice (safety), and toxicity. These dimensions were developed based on clinical literature and expert consultation, truly capturing the core of mental health QA.
- **Innovative Adversarial Benchmark**: The second component, COUNSELBENCH-ADV, is highly innovative. It moves beyond passive evaluation by analyzing failure modes identified in COUNSELBENCH-EVAL (e.g., GPT-4 suggesting specific therapy techniques, Llama-3.3 being judgmental, and Gemini-1.5-Pro being apathetic) . Experts then authored 120 new questions specifically designed to trigger these identified failures, creating a systematic way to actively stress-test models.
- **Key Falsification of "LLM-as-Judge"**: As the "LLM-as-Judge" paradigm gains popularity, this paper provides a powerful demonstration of its limitations in high-stakes domains. The study found that LLM judges "systematically overrate model responses" and "overlook safety concerns identified by human experts". For instance, LLM judges failed to identify most toxic content or factual errors. This finding serves as a critical warning for the field.

**Weaknesses:**

- **Representativeness of the "Human Baseline"**: This is a limitation of the data source, not the study's methodology. The "human therapist" baseline was sourced from the "top-voted answer" for each question on CounselChat. The authors acknowledge that these "forum contributions are informal and vary in quality". Curiously, LLaMA-3.3 outperformed this human baseline on five of the six dimensions. This makes the results hard to interpret: have LLMs surpassed human performance, or have they simply surpassed a (potentially low-quality) standard of informal forum answers?
- **Limitation of a Single Data Source**: All patient questions were sourced from a single public forum, CounselChat. The style of questions and needs expressed in this asynchronous, public format may differ significantly from those in private therapeutic conversations, EHR messages, or peer-support forums.
- **Reliance on LLM-as-Judge in Adversarial Evaluation**: This presents a core methodological contradiction. In COUNSELBENCH-EVAL, the paper demonstrates that LLM-as-Judge is unreliable. However, to assess the 3,240 responses in COUNSELBENCH-ADV, the study used GPT-4.1 as a "scalable labeling tool". While the authors validated this as a "practical proxy" via a human spot-check (72.9% agreement), this reliance undoubtedly weakens the reliability of the adversarial results.

**Questions:**

In the first part of your paper, you convincingly demonstrated that LLM judges "systematically overrate" responses in the mental health domain. Why, then, did you choose to use GPT-4.1 as the primary evaluator for the 3,240 responses in COUNSELBENCH-ADV? Despite the 72.9% agreement, does this not imply that the failure rates in Table 3 might be underestimated, given that the evaluation paradigm itself was one you proved to be unreliable?

---

> ### Author Response · Authors · 2025-11-21
> **Response to W1**
>
> Thank you for your thoughtful feedback. We appreciate the recognition of our work's quality and significance. We respond to each point below:
>
>
> 1. Representativeness of the "Human Baseline": This is a limitation of the data source, not the study's methodology. The "human therapist" baseline was sourced from the "top-voted answer" for each question on CounselChat. The authors acknowledge that these "forum contributions are informal and vary in quality". Curiously, LLaMA-3.3 outperformed this human baseline on five of the six dimensions. This makes the results hard to interpret: have LLMs surpassed human performance, or have they simply surpassed a (potentially low-quality) standard of informal forum answers?
>
> Thank you for this comment. First, we want to quickly clarify: CounselChat responses are written by verified, non-anonymous licensed therapists with public profiles; only the patient questions are anonymous. We have added this emphasis in Sec. 4.1 to avoid confusion:
> > All responses on CounselChat are written by verified, non-anonymous therapists with public profiles; only patient questions are anonymous [1].
>
> We conducted a brief quality check of the online therapist responses (median length 142 words, IQR: 95.75-183.5; median 372.5 views, IQR: 248.5-785.5), indicating reasonable substance and engagement.
>
> We fully agree that these online responses are not equivalent to those produced in formal clinical encounters. Due to stringent privacy protections, real patient-provider treatment data and clinically supervised counseling transcripts cannot be released publicly, and therefore cannot be used to build public benchmark datasets. Compared to general peer-support forums like Reddit, CounselChat provides comparatively higher-quality responses from verified clinicians, making it one of the best publicly accessible sources for open-ended mental-health QA. To avoid overstating the comparison, we carefully frame the responders as “online human therapists” throughout the paper (rather than general "human" or “human therapists”).
>
> [1] N. Bertagnolli, “Counsel Chat: Bootstrapping High-Quality Therapy Data,” Medium, May 11, 2020.

---

> ### Author Response · Authors · 2025-11-21
> **Response to W2**
>
> 2. Limitation of a Single Data Source: All patient questions were sourced from a single public forum, CounselChat. The style of questions and needs expressed in this asynchronous, public format may differ significantly from those in private therapeutic conversations, EHR messages, or peer-support forums.
>
> Thank you for raising this point. CounselChat is among the very few openly accessible resources that pair patient mental health questions with verified therapist responses, and this limitation arises from the scarcity of suitable public datasets. We have added more details in the Discussion:
>
> > As noted in prior surveys of mental‑health datasets [1, 2], publicly available data with clinician responses are extremely scarce. On one hand, most clinical conversations and patient-provider messages cannot be released due to privacy protections (e.g., de-identification risks highlighted by [3]). On the other hand, peer‑support forums [4] and social‑media platforms such as Reddit [5] typically contain non‑clinical, user‑generated content with noisy labels and without professional verification [6]. Within this landscape, CounselChat remains one of the few openly accessible resources that pair patient questions with verified therapist responses, and prior work has examined its relatively high quality and utility for research [8].
>
> Although existing public datasets are limited, the evaluation framework we develop can be directly applied to future datasets as they become available. We have added to the Discussion:
> >  Although current public datasets are limited, the evaluation framework in CounselBench, including the six-dimension rubric and annotation protocol, can be applied directly to future datasets. This structure can support consistent model comparison across broader settings in subsequent work.
>
> [1] K. Harrigian, C. Aguirre, and M. Dredze, “On the State of Social Media Data for Mental Health Research,” arXiv (Cornell University), Jan. 2021.
>
> [2] A. Mandal, A. P. Kumar, H. Arnaout, I. Gurevych, and T. Chakraborty, “A Comprehensive Review of Datasets for Clinical Mental Health AI Systems,” arXiv.org, 2025.
>
> [3] Aleksandar Kovačević, Bojana Bašaragin, Nikola Milošević, and Goran Nenadić, “De-identification of clinical free text using natural language processing: A systematic review of current approaches,” Artificial intelligence in medicine, vol. 151, pp. 102845–102845, May 2024.
>
> [4] P. Marshall et al., “Understanding the Impacts of Online Mental Health Peer Support Forums: Realist Synthesis,” JMIR Mental Health, vol. 11, no. 1, p. e55750, May 2024.
>
> [5] N. Boettcher, “Studies of depression and anxiety using Reddit as a data source: Scoping review,” JMIR Mental Health, vol. 8, no. 11, Apr. 2021.
>
> [6] M. Garg, “Mental Health Analysis in Social Media Posts: A Survey,” Archives of Computational Methods in Engineering, vol. 30, Jan. 2023.
>
> [7] N. Bertagnolli, “Counsel Chat: Bootstrapping High-Quality Therapy Data,” Medium, May 11, 2020.
>
> [8] S. Ahmed, S. Khurshid, M. Imran, Muhammad Shoaib Siddiqui, S. Hina, and M. Ahmed, “Analysis of Mental Health Counseling Conversation Using Natural Language Processing,” Journal of Computer Science, vol. 20, no. 3, pp. 303–309, Mar. 2024.

---

> ### Author Response · Authors · 2025-11-21
> **Response to W3 and Q**
>
> 3. (W3+Q) **W3**: Reliance on LLM-as-Judge in Adversarial Evaluation: This presents a core methodological contradiction. In COUNSELBENCH-EVAL, the paper demonstrates that LLM-as-Judge is unreliable. However, to assess the 3,240 responses in COUNSELBENCH-ADV, the study used GPT-4.1 as a "scalable labeling tool". While the authors validated this as a "practical proxy" via a human spot-check (72.9% agreement), this reliance undoubtedly weakens the reliability of the adversarial results. | **Q**: In the first part of your paper, you convincingly demonstrated that LLM judges "systematically overrate" responses in the mental health domain. Why, then, did you choose to use GPT-4.1 as the primary evaluator for the 3,240 responses in COUNSELBENCH-ADV? Despite the 72.9% agreement, does this not imply that the failure rates in Table 3 might be underestimated, given that the evaluation paradigm itself was one you proved to be unreliable?
>
> Thank you for raising this important concern. We agree that relying on LLM-as-Judge for evaluation would be inconsistent with our findings, and the spot-check agreement is not sufficiently high. Based on your feedback, we made a major change to the evaluation of COUNSELBENCH-ADV.
>
> **We have replaced the primary GPT-4.1 evaluation with full human annotation.** We recruited five mental-health professionals (different from those who authored the adversarial questions) to annotate 1,080 QA pairs (120 adversarial questions × 9 models × 1 response). In this setting, the task is a single binary label (whether the issue is present or not), which is fast and straightforward for experts to evaluate. Details about the evaluation setup are provided in Sec. 5 and Appendix H.
>
> The main evaluation results for COUNSELBENCH-ADV are now entirely based on human expert judgments. The updated Table 3 with expert-annotated failure rates is included in the revision:
> | Specific Issue | GPT-3.5-Turbo | GPT-4 | GPT-5 | Llama-3.1 | Llama-3.3 | Claude-3.5-Sonnet | Claude-3.7-Sonnet | Gemini-1.5-Pro | Gemini-2.0-Flash |
> |----------------|--------------:|------:|------:|----------:|----------:|-------------------:|-------------------:|---------------:|-----------------:|
> | 1. Medication  |          0.05 |  0.00 |  0.47 |      0.05 |      0.10 |               0.00 |               0.00 |           0.00 |             0.00 |
> | 2. Therapy     |          0.20 |  0.20 |  0.85 |      0.55 |      0.65 |               0.45 |               0.50 |           0.20 |             0.26 |
> | 3. Symptoms    |          0.15 |  0.45 |  0.60 |      0.45 |      0.45 |               0.50 |               0.37 |           0.26 |             0.25 |
> | 4. Judgmental  |          0.25 |  0.25 |  0.05 |      0.11 |      0.10 |               0.05 |               0.10 |           0.20 |             0.10 |
> | 5. Apathetic   |          0.70 |  0.20 |  0.15 |      0.15 |      0.15 |               0.05 |               0.20 |           0.40 |             0.30 |
> | 6. Assumptions |          0.40 |  0.35 |  0.15 |      0.25 |      0.25 |               0.35 |               0.25 |           0.40 |             0.35 |
>
> We further systematically analyze the performance of LLMs as judges for identifying failure modes. We added Table 4, which reports Accuracy and F1 of LLM judges relative to human experts:
>
> | Metrics | GPT-3.5-turbo | GPT-4 | GPT-4.1 | GPT-5 | Llama-3.1-70B-Instruct | Llama-3.3-70B-Instruct | Claude-3.7-sonnet | Gemini-2.0-flash |
> |---------|---------------:|------:|--------:|------:|-----------------------:|-----------------------:|------------------:|-----------------:|
> | Acc.    |           0.74 |  0.70 |    0.64 |  0.67 |                  0.63  |                  0.64  |              0.70 |             0.63 |
> | F1      |           0.41 |  0.35 |    0.49 |  0.49 |                  0.48  |                  0.48  |              0.50 |             0.46 |
>
> (*Gemini-1.5-Pro and Claude-3.5-Sonnet, which were included in the LLM-as-judge analysis in CounselBench-EVAL, were not longer available when we ran the LLM-as-judge experiments for CounselBench-ADV)
>
> We again observe a substantial gap between LLM judgments and human experts on COUNSELBENCH-ADV. The performance breakdown of each LLM judge for each failure mode is present in Figure 8. This reinforces that LLM-as-Judge is unreliable in this domain and highlights the importance of the fully human-annotated results now presented.
>
> We have incorporated these changes and the full updated analysis in Sec. 5 of the revision.

---

### Official Review · Reviewer_GPaF · 2025-10-31

**Soundness:** 3
**Presentation:** 2
**Contribution:** 3
**Rating:** 6
**Confidence:** 3

**Summary:**

Evaluating LLMs on authentic, open-ended mental-health questions is urgent, since even small errors can cause immediate harm, but current resources lean on MCQ proxies, small expert panels, or LLM-as-judge protocols that often miss clinically salient failures, motivating a large-scale, clinician-grounded benchmark with targeted stress tests. The paper introduces CounselBench, built around a clinically grounded six-dimension rubric spanning quality and safety (overall quality, empathy, specificity, factual consistency, medical advice, toxicity).  It releases CounselBench-eval (2,000 evaluations by 100 licensed professionals on real patient queries, with span-level annotations and rationales) and CounselBench-adv (120 clinician-authored adversarial questions), shifting evaluation beyond factual correctness toward clinically meaningful behavior in mental-health QA.

**Strengths:**

1. Clear, practice-driven motivation with a clinician-grounded benchmark and stress tests. The paper targets authentic, open-ended mental-health QA, where small errors can harm users, and answers with a two-part benchmark co-designed with clinicians, including an adversarial set authored by 10 licensed professionals from failure modes observed in practice.
2. Comprehensive evaluation and careful curation. It defines a six-dimension rubric (overall quality, empathy, specificity, factual consistency, medical advice, toxicity) and scales to 2,000 expert evaluations by 100 licensed professionals with span-level annotations and written rationales, across 100 real patient questions spanning 20 topics, yielding both breadth and clinically meaningful granularity.
3. Thorough experimental coverage and transparent methodology. The work contrasts human and LLM judges under the same rubric, probes safety with a dedicated adversarial set, and documents the pipeline with an accompanying repo, enabling reproducibility.

**Weaknesses:**

1. The evaluation is static and single-turn: models answer a one-shot patient question with no opportunity to ask clarifying questions, adapt to feedback, or repair errors across turns, capabilities that are central to safe, supportive counseling dialogues.
2. The adversarial component is also single-round and non-iterative: prompts are issued once (with three stochastic samples) and failure triggers are labeled post hoc (largely by GPT-4 with limited human validation), rather than via an attacker–defender loop that escalates difficulty across rounds and model updates. This design surfaces important vulnerabilities but underestimates the challenges of sustained red-teaming and recovery dynamics.

**Questions:**

See weakness

---

> ### Author Response · Authors · 2025-11-21
> **Response to W1 and W2**
>
> Thank you for your helpful feedback. We appreciate the recognition of our contribution and address each point below:
>
>
> 1. W1: The evaluation is static and single-turn. W2: The adversarial component is also single-round and non-iterative.
>
> Thank you for these thoughtful questions. We fully agree that conversational and dialogue-based interactions are important in mental-health support. We also note that **open-ended QA is itself an important setting with clear value for both LLM evaluation and real-world application**:
>
> First, QA is a standard evaluation paradigm for LLMs. In traditional QA, a system receives a stand-alone question and produces an answer based only on that input. This provides a controlled and reproducible setup where all models are compared on identical questions without dialogue-history confounds. We added the following clarification to the Introduction:
> > We evaluate models in an open-ended QA framework, a standardized setting that enables consistent comparison of model responses to identical patient questions.
>
> Second, single-turn interaction is also a setting of standalone clinical and practical importance. To make this clear, we expanded the following passage in the Introduction:
> > Many real-world mental health platforms, such as CounselChat [1], peer-support forums [2], EHR-based messaging, and digital services like NOCD [3], primarily operate through single-turn interactions, with limited follow-up in most cases. These settings are designed for brief, asynchronous exchanges that offer emotional support, psychoeducation, or coping strategies in a concise format. Such platforms are widely used and often serve as a primary source of support, especially for individuals who may not have access to or feel ready for full-length therapy [4,5].
>
> Finally, thank you for raising the point about conversational dialogue. We fully acknowledge the importance of multi-turn evaluation and see CounselBench as one basis for future extensions in this direction. We have added to the Discussion:
> > While CounselBench is designed for evaluating open-ended mental health QA, our framework provides a foundation for future expansion into dialogue settings. For CounselBench-Eval, the current six-dimension rubric can be applied turn-by-turn and aggregated across the dialogue. For example, by averaging scores or weighting them based on conversational role (e.g., first vs. follow-up turns). However, evaluating multi-turn exchanges introduces additional complexities, including the need for models to appropriately track and utilize prior conversational context. This raises challenges in both prompting design (e.g., how much history to include) and evaluation (e.g., how to measure coherence or consistency over time). For CounselBench-Adv, extending to multi-turn adversarial prompts requires preserving interactional dynamics while targeting specific failure modes. One promising direction is to use simulated patient agents to scaffold coherent multi-turn interactions that elicit subtle safety or helpfulness failures [6]. We view CounselBench as a step toward this broader goal, and hope it catalyzes future work on longitudinal evaluation in mental health dialogue.
>
> In summary, QA is an established evaluation setting with its own importance, and there is currently limited expert-annotated, open-ended mental-health QA data available. To the best of our knowledge, CounselBench is one of the largest such resources, and we hope it can also catalyze future work on longitudinal evaluation in mental-health dialogue.
>
>
> [1] N. Bertagnolli, “Counsel Chat: Bootstrapping High-Quality Therapy Data,” Medium, May 11, 2020.
>
> [2] S. R. Pendse, K. Niederhoffer, and A. Sharma, “Cross-Cultural Differences in the Use of Online Mental Health Support Forums,” Proceedings of the ACM on Human-Computer Interaction, vol. 3, no. CSCW, pp. 1–29, Nov. 2019.
>
> [3] “With NOCD Therapy, you’re never alone,” NOCD, 2018.
>
> [4] J. A. Naslund, K. A. Aschbrenner, L. A. Marsch, and S. J. Bartels, “The Future of Mental Health care: Peer-to-peer Support and Social Media,” Epidemiology and Psychiatric Sciences, vol. 25, no. 2, pp. 113–122, 2016.
>
> [5] R. Merchant et al., “Opportunities to expand access to mental health services: A case for the role of online peer support communities,” Psychiatric Quarterly, vol. 93, pp. 613–625, Feb. 2022.
>
> [6] J. Wang et al., “Towards a Client-Centered Assessment of LLM Therapists by Client Simulation,” arXiv.org, 2024.

---

> ### Author Response · Authors · 2025-11-21
> **Response to one point in W2**
>
> 2. Failure triggers are labeled largely by GPT-4 with limited human validation.
>
> Thank you for raising this important concern. We agree that relying on GPT-4.1 for failure-mode labeling is not reliable and would diminish the validity of the findings. We have made a major change to the evaluation of COUNSELBENCH-ADV.
>
> **We have replaced the primary GPT-4.1 evaluation with full human annotation.** We recruited five mental-health professionals (different from those who authored the adversarial questions) to annotate 1,080 QA pairs (120 adversarial questions × 9 models × 1 response). In this setting, the task is a single binary label (whether the issue is present or not), which is fast and straightforward for experts to evaluate. Details about the evaluation setup are provided in Sec. 5 and Appendix H.
>
> The main evaluation results for COUNSELBENCH-ADV are now entirely based on human expert judgments. The updated Table 3 with expert-annotated failure rates is included in the revision:
> | Specific Issue | GPT-3.5-Turbo | GPT-4 | GPT-5 | Llama-3.1 | Llama-3.3 | Claude-3.5-Sonnet | Claude-3.7-Sonnet | Gemini-1.5-Pro | Gemini-2.0-Flash |
> |----------------|--------------:|------:|------:|----------:|----------:|-------------------:|-------------------:|---------------:|-----------------:|
> | 1. Medication  |          0.05 |  0.00 |  0.47 |      0.05 |      0.10 |               0.00 |               0.00 |           0.00 |             0.00 |
> | 2. Therapy     |          0.20 |  0.20 |  0.85 |      0.55 |      0.65 |               0.45 |               0.50 |           0.20 |             0.26 |
> | 3. Symptoms    |          0.15 |  0.45 |  0.60 |      0.45 |      0.45 |               0.50 |               0.37 |           0.26 |             0.25 |
> | 4. Judgmental  |          0.25 |  0.25 |  0.05 |      0.11 |      0.10 |               0.05 |               0.10 |           0.20 |             0.10 |
> | 5. Apathetic   |          0.70 |  0.20 |  0.15 |      0.15 |      0.15 |               0.05 |               0.20 |           0.40 |             0.30 |
> | 6. Assumptions |          0.40 |  0.35 |  0.15 |      0.25 |      0.25 |               0.35 |               0.25 |           0.40 |             0.35 |
>
> We further systematically analyze the performance of LLMs as judges for identifying failure modes. We added Table 4, which reports Accuracy and F1 of LLM judges relative to human experts:
>
> | Metrics | GPT-3.5-turbo | GPT-4 | GPT-4.1 | GPT-5 | Llama-3.1-70B-Instruct | Llama-3.3-70B-Instruct | Claude-3.7-sonnet | Gemini-2.0-flash |
> |---------|---------------:|------:|--------:|------:|-----------------------:|-----------------------:|------------------:|-----------------:|
> | Acc.    |           0.74 |  0.70 |    0.64 |  0.67 |                  0.63  |                  0.64  |              0.70 |             0.63 |
> | F1      |           0.41 |  0.35 |    0.49 |  0.49 |                  0.48  |                  0.48  |              0.50 |             0.46 |
>
> (*Gemini-1.5-Pro and Claude-3.5-Sonnet, which were included in the LLM-as-judge analysis in CounselBench-EVAL, were not longer available when we ran the LLM-as-judge experiments for CounselBench-ADV)
>
> We again observe a substantial gap between LLM judgments and human experts on COUNSELBENCH-ADV. The performance breakdown of each LLM judge for each failure mode is present in Figure 8. This reinforces that LLM-as-Judge is unreliable in this domain and highlights the importance of the fully human-annotated results now presented.
>
> We have incorporated these changes and the full updated analysis in Sec. 5 of the revision.

---

### Author Response · Authors · 2025-12-03
**Response Summary**

Here we briefly summarize our response to the reviewers’ concern:


> **Reviewer GPaF (score 6)**

- **Weakness (2 points):** Both concerns target the use of single-turn setting. W1: “The evaluation is static and single-turn”; W2: “The adversarial component is also single-round and non-iterative”.
- **Response:** We clarify that we benchmark open-ended QA, not multi-turn dialogue. Multi-turn interaction is out of scope and should not diminish our contribution. **Open-ended QA is an important setting on its own**:

First, QA is a standard evaluation paradigm for LLMs.

Second, QA is a clinically and practically meaningful setting. We added examples where mental-health services operate primarily through single-turn QA: CounselChat, peer-support forums, EHR-based messaging, NOCD.

We further added to Discussion how our framework can be extended to dialogue settings. We emphasize that QA is an established evaluation setting with its own importance, and CounselBench is one of the largest expert-annotated, open-ended mental-health QA resources.

- **An additional point in W2:** Failure triggers (Counselbench-ADV) are labeled largely by GPT-4 with limited human validation..
- **Response:** To address the concern, we recruited five mental health professionals, and **replaced the GPT-4 evaluation with full human annotation**. In the revision, the results are now entirely based on human expert judgments. More details in Response and Revision.



> **Reviewer HDvA (score 6)**

- **W1, W2:** W1: Representativeness of the "Human Baseline", since responses come from a public forum. W2: Limitation of a Single Data Source: CounselChat.
- **Response:** First, we clarify that CounselChat responses are written by **verified, non-anonymous licensed therapists with public profiles**, not anonymous forum users.  Second, strict privacy protections make real patient-provider treatment data unavailable for public benchmarks. As a result, CounselChat is one of the few openly accessible sources that pair patient questions with verified therapist responses. We added discussion on the scarcity of suitable public datasets and note that our framework can be directly applied to future datasets.

- **W3, Q:** Reliance on LLM-as-Judge in Adversarial Evaluation.
- **Response:** (Same concern for reviewer GPaF.) We recruited five mental health professionals, and **replaced the LLM-as-Judge with full human annotation**. In the revision, the results are now entirely based on human expert judgments. More details in Response and Revision.



> **Reviewer eiyu (score 8)**

- **W1, Q1:** Limited size and diversity of 100 questions.
- **Response:** We note that evaluation is at the QA-pair level, yielding 400 distinct QA pairs and 2,000 expert annotations. We also added Table 13 to summarize keywords for questions across 20 categories, demonstrating substantial within-topic diversity.

- **W2, Q3:** Heavy reliance on automated evaluation (LLM-as-Judge) of the adversarial responses.
- **Response:** (Same concern for the last two reviewers.) We recruited five mental health professionals, and **replaced the automated evaluation with full human annotation**. In the revision, the results are now entirely based on human expert judgments. More details in Response and Revision.

- **W3, Q2, Q4:** The adversarial benchmark is confusing. Q2: What makes the questions adversarial? Q4: Should we expect a question designed to trigger specific vulnerabilities to also trigger issues across model families?
- **Response:** We added more clarification of the construction of adversarial benchmark in the revision. First, our goal is to recreate realistic questions that reliably elicit specific failure modes observed in Counselbench-EVAL. Second, our method follows established approaches in adversarial data generation. Finally, we clarify three aspects regarding whether these failure modes are model-specific (details in Response).

- **W4:** Is it truly fair to benchmark against the human baselines as they were often informal and vary in quality as they could be answered anonymously.
- **Response:** We clarify that CounselChat responses are written by **verified, non-anonymous licensed therapists with public profiles**, not anonymous forum users.

---

### Meta-Review · Area_Chair_nZBQ · 2025-12-17

**Summary:**

This paper presents CounselBench, a comprehensive benchmark for evaluating the ability of LLMs in open-ended mental health QA. Developed with input from 100 licensed mental health professionals, CounselBench consists of two components: CounselBench-EVAL, featuring 2,000 expert evaluations of responses to real patient questions across six clinically relevant dimensions, and CounselBench-Adv, which includes 120 adversarial prompts intended to uncover specific failure modes. While LLMs demonstrate high scores in certain quality dimensions, they often struggle with issues like personalization and safety, highlighting the need for clinically grounded evaluation frameworks. This research aims to enhance the understanding of LLM performance in sensitive contexts and foster improvements in their responses.

**Reviewer Concerns:**

Most of the weaknesses that were raised by reviewers were addressed by the authors. More specifically:

- *Static and Single-Turn Evaluation*: Reviewers wished authors had employed a more dynamic evaluation methodology that allowed for clarifying questions and iterative dialogue, crucial for safe counseling interactions. **Authors argued that open-ended single-turn QA is still very useful resource, and the AC agrees.**

- *LLM-as-a-judge reliance*: Two reviewers noted the adversarial component was limited to single-round assessments and lacked iterative engagement, which could result in undervaluing the complexity of red-teaming processes. **Authors agreed and addressed this by replacing GPT-4.1 evaluation with full human annotation conducted by mental health professionals, while also benchmarking LLM-judges against human evaluations, finding that GPT-3.5-Turbo is the best model.**

- *Concerns About Human Baseline Representativeness*: Reviewers felt that the quality of human responses used as a baseline may not provide a reliable standard given their informal nature, complicating the interpretation of LLaMA-3.3's performance exceeding this baseline. **Authors clarified that CounselChat responses are generated by verified, licensed therapists but should further explain why LLaMA responses may outperform human responses in the camera-ready version.**

- *Dataset Limitations*: Reviewers highlighted the limited scale and diversity of the dataset, with only 100 patient questions and 120 adversarial benchmarks, which might not adequately capture the complexity of mental health issues and interactions. **Authors acknowledged this limitation, though the AC and reviewers did not believe it was a major concern.**

- *Adversarial Benchmark Clarity*: A reviewer pointed out confusion in defining the nature of adversarial questions versus typical difficult questions and questioned the universality of the identified issues across different models. **Authors adequately addressed this by clarifying what constitutes an adversarial question.**

**Reviewer Scores:**

I thikn the scores were already pretty high but yes I think so.

---

### Decision · Program_Chairs · 2026-01-26

Accept (Oral)